# Amelioration of Murine Colitis by Attenuated *Salmonella choleraesuis* Encoding Interleukin-19

**DOI:** 10.3390/microorganisms11061530

**Published:** 2023-06-08

**Authors:** Shih-Yao Chen, Chun-Ting Chu, Mei-Lin Yang, Jian-Da Lin, Chung-Teng Wang, Che-Hsin Lee, I-Chen Lin, Ai-Li Shiau, Pin Ling, Chao-Liang Wu

**Affiliations:** 1Department of Nursing, College of Nursing, Chung Hwa University of Medical Technology, Tainan 71703, Taiwan; leonlai50@gmail.com; 2Division of Colorectal Surgery, Department of Surgery, Ditmanson Medical Foundation Chia-Yi Christian Hospital, 539, Zhongxiao Road, Chiayi City 60002, Taiwan; 3Ditmanson Medical Foundation Chia-Yi Christian Hospital, Chiayi City 60002, Taiwan; 4Department of Microbiology and Immunology, College of Medicine, National Cheng Kung University, Tainan 70101, Taiwan; 5Department of Biochemical Science and Technology, College of Life Science, National Taiwan University, Taipei City 10617, Taiwan; 6Department of Biological Sciences, National Sun Yat-sen University, Kaohsiung 80424, Taiwan; 7Department of Biochemistry and Molecular Biology, College of Medicine, National Cheng Kung University, Tainan 70101, Taiwan

**Keywords:** inflammatory bowel diseases, interleukin-19, *Salmonella choleraesuis*, dextran sulfate sodium-induced colitis

## Abstract

The imbalance of mucosal immunity in the lower gastrointestinal tract can lead to chronic inflammatory bowel diseases (IBDs), including Crohn’s disease and ulcerative colitis. IBD is a chronic inflammatory disorder that causes small and/or large intestines ulceration. According to previous studies, recombinant interleukin (IL)-10 protein and genetically modified bacteria secreting IL-10 ameliorate dextran sulfate sodium (DSS)-induced colitis in mice. IL-19 is a transcriptional activator of IL-10 and can alter the balance of T helper 1 (Th)1/Th2 cells in favor of Th2. In this study, we aimed to investigate whether the expression of the murine IL-19 gene carried by *Salmonella choleraesuis* (*S. choleraesuis*) could ameliorate murine IBD. Our results showed that the attenuated *S. choleraesuis* could carry and express the IL-19 gene-containing plasmid for IBD gene therapy by reducing the mortality and clinical signs in DSS-induced acute colitis mice as compared to the untreated ones. We also found that IL-10 expression was induced in IL-19-treated colitis mice and prevented inflammatory infiltrates and proinflammatory cytokine expression in these mice. We suggest that *S. choleraesuis* encoding IL-19 provides a new strategy for treating IBD in the future.

## 1. Introduction

Uncontrolled mucosal immunity in the gastrointestinal tract of humans results in chronic inflammatory bowel diseases (IBD), including Crohn’s disease (CD) and ulcerative colitis (UC) [1]. IBD is a chronic inflammatory disorder in the small and/or large intestines with ulceration. Genetic, environmental, and immunologic factors can contribute to the disease. However, the exact etiology of IBD remains unclear. In general, inappropriate inflammatory responses to unknown antigens presented by the normal bacterial microflora can be found in the gut of IBD patients [2].

Interleukin (IL)-10 is a pivotal immunoregulatory cytokine that limits and terminates the inflammatory response. Studies of IL-10-deficient (IL-10^−/−^) and IL-10 receptor-2-deficient mice show that these mice develop a T helper1 (Th)1 intestinal inflammation, indicating that IL-10 can maintain gastrointestinal mucosal homeostasis [3,4,5]. IL-10 gene delivery can prevent trinitrobenzene sulfonic acid (TNBS) and dextran sodium sulfate (DSS)-induced murine colitis [4,6,7]. Recombinant IL-10 protein and genetically modified bacteria secreting IL-10 also ameliorate the disease [8,9]. Nevertheless, the mechanism underlying the IL-10-mediated amelioration of intestinal inflammation would likely be multifactorial. For instance, it is a potent inhibitor of IL-12 that limits Th1 cell induction [10]. In addition, IL-10 suppresses the production of proinflammatory cytokines and chemokines, including tumor necrosis factor-α (TNF)-α, IL-1β, IL-6, and IL-8 [11]. More importantly, strong evidence indicates that IL-10 can promote the differentiation and activity of regulatory T cells [12].

IL-19 belongs to the IL-10 cytokine family, which influences the maturation of T cells and alters the imbalance of Th1/Th2 cells in favor of Th2 [13]. Furthermore, IL-10 is strongly induced in IL-19-stimulated peripheral blood mononuclear cells (PBMCs), indicating that IL-19 is a transcriptional activator of IL-10 [14]. Therefore, it may suggest that IL-19 gene delivery can ameliorate murine colitis.

*Salmonella typhimurium* (*S. typhimurium*), a facultative anaerobe, has been developed as a gene delivery vector. It can transfer eukaryotic expression plasmids to mammalian cells in vitro and in vivo [15]. The oral administration of attenuated *S. typhimurium* carrying a eukaryotic expression plasmid encoding interferon-γ gene restores the production of this cytokine in macrophages of interferon-γ-deficient mice [16]. *S. typhimurium* harboring the plasmids with eukaryotic expressible IL-12 or granulocyte/macrophage colony-stimulating factor (GM-CSF) genes exerts anti-tumor effects in mice [17].

We have reported that attenuated *Salmonella choleraesuis* (*S. choleraesuis*) can serve as a DNA vaccine against pseudorabies and enhance both humoral and cellular immune responses [18,19]. Therefore, we planned to exploit *S. cholersesuis* carrying a eukaryotic expression plasmid encoding the IL-19 gene as a new gene therapeutic strategy in a DSS-induced murine colitis model. Through the induction of IL-10, *S. cholersesuis* carrying the IL-19 gene may have therapeutic potentials for treating murine colitis.

## 2. Material and Methods

### 2.1. Isolation of Murine IL-19 Complementary DNA (cDNA)

RAW 264.7 cells were seeded at a density of 5 × 10^6^ cells into 6-well plates. After 24 h, the cells were treated with lipopolysaccharide (LPS) (100 ng/mL) for 4 h. The total RNA was isolated with TRIzol reagent (Cat.#15596026, ThermoFisher Scientific, Waltham, MA, USA) and the cDNA was synthesized using a Verso cDNA synthesis kit (Cat.#AB1453A, ThermoFisher Scientific). Reverse transcription polymerase chain reaction (RT-PCR) was performed with the murine IL-19 specific primers (Forward: 5′-AAGGATCCATGAAGACACAGTGCG-3′; Reverse: 5′-AAGAATTCCGTTTCGTGTCAGGCT-3′) and the amplified PCR fragment (540 b.p.) was cloned into pGEM-T Easy vector (Promega, Madison, WI, USA), designated TA-mIL-19 to verify the correctness via DNA sequencing.

### 2.2. Plasmids and Bacteria

Murine IL-19 cDNA was excised from TA-mIL-19 with *Bam*HI/*Not*I sites and cloned into pEGFP-N1 (Invitrogen, Waltham, MA, USA) to replace the EGFP gene, designated pLJD-CMV-mIL19. Meanwhile, the murine IL-19 cDNA was also excised from TA-mIL19 with *Hind*III/*Bam*HI sites and cloned into pEGFP-N1, designated pLJD-CMV-mIL19-EGFP. The luciferase gene was cloned into pEGFP-N1 with *Hind*III/*Not*I sites to replace the EGFP gene, designated pLJD-CMV-Luc. The pLJD plasmid was constructed from pEGFP-N1 by releasing the EGFP gene with *Eco*RI sites and self-ligation. *S. choleraesuis*, obtained from Bioresources Collection and Research Center (Hsinchu, Taiwan) [18], was transformed with the pLJD-CMV-Luc, pLJD-CMV-mIL19-EGFP, pLJD, and pLJD-CMV-mIL19 plasmids using electroporation to obtain *S.C.*/pLJD-CMV-Luc, *S.C.*/pLJD-CMV-mIL19-EGFP, *S.C.*/pLJD, and *S.C.*/pLJD-CMV-mIL19.

### 2.3. Cell Lines and Mice

The human colon epithelial cell, Caco-2, was cultured in Dulbecco’s modified Eagle medium (DMEM) supplemented with 50 μg/mL gentamicin, 2 mM _L_-glutamine, and 20% fetal bovine serum (Hyclone, Logan, Utah, USA) at 37 °C and 5% CO_2_. The murine macrophage, RAW 264.7, was cultured in DMEM supplemented with 50 μg/mL gentamicin, 2 mM _L_-glutamine, and 10% cosmic calf serum (Hyclone, Logan, Utah, USA) at 37 °C and 5% CO_2_.

Female C57BL/6 mice (8 to 10 weeks old) were obtained from the Laboratory Animal Center of National Cheng Kung University. The animals were maintained in specific pathogen-free animal care facilities under isothermal conditions with regular photoperiods. The experimental protocol adhered to was approved by the Laboratory Animal Care and Use Committee of National Cheng Kung University (Approval numbers: 109160 and 108324).

### 2.4. Assays of Gene Transfer In Vitro and In Vivo

Caco-2 cells (5 × 10^5^/well) were cultured into 6-well plates for 24 h. Thereafter, *S.C*/pLJD-CMV-Luc and S.C. at multiplicity of infections (MOIs) of 0.1, 1, and 10 were added to cells and cultured with 1 mL of antibiotic-free medium and incubated for 8 h. After 8 h, the cells were washed, replenished with gentamycin (50 ug/mL)-containing complete medium, and cultured for 24 h. The cells were lysed to prepare extracts for the determination of the luciferase activity by a luciferase assay kit (Tropix, Bedford, MA, USA).

Mice were orally administrated with 10^8^ colony-forming units (CFUs) of *S.C.*/pLJD-CMV-mIL19-EGFP, and the colons were collected 24 h later. The colons were embedded in OCT compound (Sakura Finete U.S.A. Inc. Tissue-Tek, Torrance, CA, USA) and stored at −80 °C until further processing for immunohistochemical (IHC) staining.

### 2.5. IHC Staining

The colons were isolated from the saline, 10^8^ CFU of *S.C.*-, *S.C.*/pLJD-, or *S.C.*/pLJD-CMV-mIL19-treated DSS colitis mice at day 5 and rinsed with chilled phosphate-buffered saline (PBS). Tissues were embedded in OCT compound, serial transverse, and 5-μm-thick sections of the colons were fixed in 4% formaldehyde, incubated in cold acetone and in 3% H_2_O_2_ to block the endogenous peroxidase activity. Each section was incubated with the primary antibodies against Mac-3 (Cat.# 553322, BD Biosciences, Franklin Lakes, NJ, USA, 1:100), Gr-1 (Cat.# 553123, BD Biosciences, 1:100), and GFP (Cat.#sc-9996, Santa Cruz Biotechnology, Dallas, TX, USA) at 4 °C overnight, followed by the secondary antibodies conjugated with horseradish peroxidase (HRP) (Jackson, West Grove, PA, USA) (1:500). AEC substrate solution (Zymed Labrotary, South San Francisco, CA, USA) was used as a substrate chromogen and each section was further counterstained with hematoxylin (DakoCytomation, Glostrup, Denmark).

### 2.6. Isolation of Spleen Cells

Murine splenocytes were prepared from the spleen of 8-10-wk-old female mice. The spleen cells were cultured in Roswell Park Memorial Institute (RPMI)-1640 medium supplemented with 50 μg/mL gentamicin, 2 mM _L_-glutamine, 50 μM 2-mercaptoethanol (Sigma, St. Louis, MO, USA), 10 μM Sodium Pyruvate, non-essential amino acid and 10% fetal bovine serum (Hyclone, Logan, Utah, USA) at 37 °C and 5% CO_2_.

### 2.7. Induction of Colitis and Injection of S.C. in DSS-Induced Colitis Mice

Dextran sodium sulfate (DSS) [M.W. 36,000–50,000 (MP Biomedicals, LLC, Eschwege, Germany)] was added to tap water at a concentration of 2% for C57BL/6 mice. Fresh DSS solution was prepared daily. Mice were exposed to 2% DSS for six days. Healthy control mice received tap water only.

Mice receiving DSS were challenged with saline, 10^8^ CFU of *S.C.*, *S.C.*/pLJD, or *S.C.*/pLJD-CMV-mIL19 per mouse from day 0 to day 5 (6 doses). The schedule of the treatment is shown in Figure 1. The parameters, including survival, body weight, stool consistency, fecal bleeding, and diarrhea, were used to evaluate clinical signs in the saline, *S.C.*-, *S.C.*/pLJD-, or *S.C.*/pLJD-CMV-mIL19-treated DSS colitis mice.

### 2.8. TNF-α, IL-6, IL-1β, and IL-10 Concentration in the Colonic Tissue

The colons were removed from the saline-, *S.C.*-, *S.C.*/pLJD-, or *S.C.*/pLJD-CMV-mIL19-treated DSS colitis-treated mice at day 3 and 5. After diarrhea and fecal blood analyses, the specimens were subjected to PBS supplemented with complete miniprotease inhibitor cocktails (Roche Molecular Biochemicals, Mannheim, Germany). After homogenization, the samples were centrifuged at 16,000× *g* for 10 min at 4 °C to precipitate the insoluble cellular debris, and the supernatant was stored at −80 °C until analysis. The concentrations of TNF-α, IL-10, IL-6, and IL-1β were determined using the mouse Quantikine enzyme-linked immunosorbent assay (ELISA) kits (TNF-α and IL-10, Preprotech) (IL-6 and IL-1β, R&D Systems, Minneapolis, MN, USA).

### 2.9. Histological Evaluation and Scoring

The colons were isolated from the mice and assessed for diarrhea and fecal blood analyses by the blind pathophysiologists. Then, the colon specimens were fixed in 10% paraformaldehyde, embedded in paraffin and sliced into sections. The sections were stained with haematoxylin and eosin (H&E), and histological analysis was performed in a blinded fashion. The pathophysiology of the tissue was characterized by the presence of ulceration, inflammatory cells (neutrophils, macrophages, lymphocytes, and plasma cells), signs of edema, crypt loss, surface epithelial cell hyperplasia, goblet cell reduction, and signs of epithelial regeneration. Diarrhea score, fecal blood score, and inflammatory score were evaluated as previously described [20].

### 2.10. Statistical Analysis

The data are expressed as the mean ± SEM. The differences between the two groups and among the groups were analyzed using Student’s *t*-test and one-way ANOVA followed by Dunnet multiple comparison tests, respectively. Differences in body weight changes were compared using repeated-measures analysis of variance. The percent survival was compared using a log-rank (Mantel–Cox) test (Prism 5.0). *p*-values less than 0.05 were considered significant.

## 3. Results

### 3.1. S. choleraesuis Delivered Reporter Genes In Vitro and In Vivo

To examine the ability of *S. choleraesuis* to transfer plasmid in vitro, we infected Caco-2 cells with *S. choleraesuis* carrying the eukaryotic expression plasmid pLJD-CMV-Luc (*S.C.*/pLJD-CMV-Luc). Caco-2 cells were incubated with *S.C.*/pLJD-CMV-Luc for 8 h or with *S.C.* only. Monolayers were then washed and cultured in a complete medium for 24 h, and the luciferase activity was determined. Figure 2a shows that the luciferase activity gradually increases in Caco-2 cells infected with *S.C.*/pLJD-CMV-Luc, but not with *S.C.* only. To investigate the ability of *S. choleraesuis* to transfer genes to the gastrointestinal tract, we injected mice with 10^8^ CFU of *S.C.*/pLJD-CMV-mIL19-GFP. Immunohistochemical staining showed the GFP proteins were located at the intestinal glands, lamina propria, and muscular mucosa (Figure 2b), suggesting that *S. choleraesuis* could help to deliver the eukaryotic plasmid in vitro and to the mouse gastrointestinal tract in vivo.

### 3.2. S.C./pLJD-CMV-mIL19-Mediated Gene Transfer Up-Regulated IL-10 Expression In Vitro and In Vivo

IL-19 is a transcriptional activator of IL-10 in PBMCs [14]. To examine whether the *S.C.*/pLJD-CMV-mIL19 can express IL-19 and transactivates IL-10 in vitro, Caco-2 cells were infected with *S.C.*/pLJD-CMV-mIL19 for 24 h, and the conditioned media were collected to treat the murine macrophages RAW 264.7 and spleen cells. The IL-10 concentration in RAW 264.7 treated with conditioned media from *S.C.*/pLJD-CMV-mIL19-infected Caco-2 (2988.51 pg/mL) was higher than those treated with Caco-2 medium alone (175.41 ± 9.5 pg/mL, *p* < 0.001), *S.C.*-infected Caco-2 conditioned media (1980.99 ± 93 pg/mL, *p* = 0.001), and *S.C.*/pLJD-infected Caco-2 conditioned media (1943.49 ± 69 pg/mL, *p* < 0.001) (Figure 3a). Furthermore, we isolated splenocytes from C57BL/6 mice and treated them with various Caco-2 conditioned media for 24 h, as described above. The IL-10 concentration in splenocytes treated with conditioned media from *S.C.*/pLJD-CMV-mIL19-infected Caco-2 (17.11 ± 2.23 pg/mL) was also higher than those treated with Caco-2 media alone (1.72 ± 1.72 pg/mL, *p* = 0.005), *S.C.*-infected Caco-2 conditioned media (1.72 ± 1.72 pg/mL, *p* = 0.005), and *S.C.*/pLJD-infected Caco-2 conditioned media (0.24 ± 0.24 pg/mL, *p* = 0.001) (Figure 3b). To examine if *S.C.*/pLJD-CMV-mIL19 could induce IL-10 in vivo, colitis was induced in C57BL/6 mice receiving 2% DSS for 3 days, and 10^8^ CFU of *S.C.*/pLJD-CMV-mIL19 was delivered orally (day 0–3) into the gastrointestinal tract of DSS-induced mice. DSS-induced control mice received 10^8^ CFU of *S.C.*, *S.C.*/pLJD, or saline alone. At day 3, IL-10 and IL-19 expression levels were increased in *S.C.*/pLJD-CMV-mIL19-treated DSS colitis mice compared to control mice, as determined by RT-PCR (Figure 4a). Furthermore, the IL-10 protein levels in colons from *S.C.*/pLJD-CMV-mIL19-treated DSS mice (21.3 ± 0.49 ng/100 mg tissue) were higher than those from DSS mice treated with saline (14.0 ± 0.49 ng/100 mg tissue, *p* < 0.001), *S.C.* (12.1 ± 0.63 ng/100 mg tissue, *p* < 0.001), and *S.C.*/pLJD (10.6 ± 0.46 ng/100 mg tissue, *p* < 0.001) (Figure 4b).

### 3.3. S.C./pLJD-CMV-mIL19 Treatment Reduced Mortality and Body Weight Loss in DSS-Induced Acute Colitis Mice

To determine whether *S.C.*/pLJD-CMV-mIL19 could prevent acute colitis, C57BL/6 mice (total of 10 mice per group) receiving 2% DSS for six days were orally delivered 10^8^ CFU of *S.C.*/pLJD-CMV-mIL19 (day 0–5) into the gastrointestinal tracts of the mice. Control mice with DSS-induced colitis received 10^8^ CFU of *S.C.*, *S.C.*/pLJD, or saline alone. At day 5, the control mice showed significant body weight loss (saline, *p* = 0.001; *S.C.*, *p* = 0.003; *S.C.*/pLJD, *p* = 0.002 in comparison to *S.C.*/pLJD-CMV-IL19 treatment) (Figure 5a). On the other hand, *S.C.*/pLJD-CMV-mIL19-treated colitis mice suffered from less weight loss than control mice (Figure 5a). There was a 40% mortality in the *S.C.*/pLJD-CMV-mIL19-treated mice when compared to the control mice, with 100% mortality at day 11 (Figure 5b). These results indicated that the oral administration of *S.C.*/pLJD-CMV-mIL19 significantly attenuated body weight loss and reduced mortality in DSS-induced colitis mice.

### 3.4. S.C./pLJD-CMV-mIL19 Ameliorated Clinical Signs in the DSS-Induced Colitis Mice

After sacrifice, the excised colons were evaluated for diarrhea and fecal blood, and the assessments were assessed by blind histopathologists, as described previously [20]. The control mice showed significantly higher diarrhea and fecal blood score than *S.C.*/pLJD-CMV-mIL19-treated mice at day 5 (diarrhea score: saline, *p* = 0.0008, *S.C.*, *p* = 0.0004, *S.C.*/pLJD, *p* = 0.0001; fecal blood score: saline, *S.C.*, and *S.C.*/pLJD, *p* = 0.001 in comparison to *S.C.*/pLJD-CMV-mIL19 treatment) (Figure 5c,d). Thus, these findings indicate that *S.C.*/pLJD-CMV-mIL19 therapy diminishes both diarrhea and fecal blood inthe DSS acute colitis mice.

### 3.5. S.C./pLJD-CMV-mIL19 Decreased Intestinal Inflammation in DSS-Induced Acute Colitis Mice

At day 3, the control mice showed minimal changes in the surface epithelium and slight infiltration of inflammatory cells to the mucosa. At day 5, there were changes in the control mice, with a loss of crypts and a reduction in goblet cells, signs of surface epithelial regeneration, focal ulcerations, moderate infiltration of inflammatory cells to the mucosa, and edema in the submucosa. *S.C.*/pLJD-CMV-mIL19-treated mice showed normal symptoms of mucosa but a slight infiltration of inflammatory cells to the mucosa at day 5 (Figure 6a). At day 5, the control mice showed significantly higher inflammatory scores than the *S.C.*/pLJD-CMV-mIL19-treated mice (saline, *p* = 0.007, *S.C.*, *p* = 0.01, *S.C.*/pLJD, *p* = 0.01 in comparison to *S.C.*/pLJD-CMV-IL19 treatment (Figure 6b). Furthermore, control mice had moderate infiltration of Mac-3- and Gr1-positive cells when compared to *S.C.*/pLJD-CMV-mIL19 treated mice in the mucosa (Figure 7). Taken together, *S.C.*/pLJD-CMV-IL19 treatment can reduce macrophage and neutrophil infiltrations to the mucosa in DSS-induced acute colitis mice.

### 3.6. S.C./pLJD-CMV-mIL19 Treatment Decreased Colonic Proinflammatory Cytokine Production

To investigate whether the therapeutic effect of *S.C.*/pLJD-CMV-mIL19 was attributed to the reduced proinflammatory cytokines expression, we analyzed the levels of TNF-α, IL-6, IL-1β in colonic tissue homogenates. At day 3, TNF-α and IL-1β levels from *S.C.*/pLJD-CMV-mIL19-treated DSS mice were significantly lower than those from DSS mice treated with saline (5.67 ± 0.18 and 1.99 ± 0.05 ng/100 mg tissue, respectively, *p* < 0.001), *S.C.* (2.26 ± 0.49 ng/100 mg tissue, *p* = 0.09 and 1.68 ± 0.03 ng/100 mg tissue, respectively, *p* < 0.001), and *S.C.*/pLJD (1.91 ± 0.19 and 1.29 ± 0.05 ng/100 mg tissue, respectively, both of *p* < 0.001) (Figure 8a,b). At day 5, TNF-α, IL-6, and IL-1β concentration from *S.C.*/pLJD-CMV-mIL19-treated DSS mice (0.25 ± 0.06 ng/100 mg tissue, under detection, and 1.35 ± 0.14 ng/100 mg tissue, respectively) were lower than those from DSS mice treated with saline (7.74 ± 1.37 ng/100 mg tissue, *p* = 0.005; 12.9 ± 2.90 ng/100 mg tissue, *p* = 0.011 and 10.3 ± 0.31 ng/100 mg tissue, *p* < 0.01, respectively), *S.C.* (2.71 ± 0.88 ng/100 mg tissue *p* = 0.04; 3.58 ± 0.44 ng/100 mg tissue, *p* = 0.001 and 6.47 ± 0.29 ng/100 mg tissue, *p* < 0.01, respectively), and *S.C.*/pLJD (4.30 ± 0.83 ng/100 mg tissue *p* = 0.008; 2.36 ± 0.62 ng/100 mg tissue, *p* < 0.001 and 5.52 ± 0.45 ng/100 mg tissue, *p* < 0.001, respectively) (Figure 8c–e). These results indicated that *S.C.*/pLJD-CMV-mIL19 reduced colon inflammation by modulating the production of proinflammatory cytokines such as TNF-α, IL-6, and IL-1β. 

## 4. Discussion

The transfer of bacteria-mediated eukaryotic expression plasmids to mammalian cells has been applied to gene therapy. Bacteria, such as *S. typhimurium*, *S. typhi* or *E. coli*, invade their host cells by expressing the invasion of *Y. pseudotuberculosis* and then remain in the vacuole. Due to metabolic reduction, they die there and release their expression plasmids. With the unknown mechanisms, the plasmids can cross the vesicular membrane and translocate into the cell nucleus of the host cells, and they are expressed there [15]. According to these findings, we showed that *S. choleraesuis* transferred eukaryotic expression plasmids into colon epithelial cells in vitro, resulting in transgene expression in a dose-dependent manner (Figure 2a). Furthermore, we also showed that *S. choleraesuis* transferred eukaryotic expression plasmids into the intestinal glands, lamina propria and muscular mucosa (Figure 2b). Thus, we designated an intracellular bacteria *S. choleraesuis* carrying a eukaryotic-expressing plasmid encoding IL-19 to evaluate the therapeutic effect on murine colitis.

IL-19-deficient mice suffered from exacerbated experimentally induced colitis with the increased production of several proinflammatory cytokines, indicating IL-19 to be an anti-inflammatory cytokine in intestinal inflammation in mice [21,22]. To our knowledge, this should be the first report revealing murine IL-19 gene therapy via attenuated *S. choleraesuis* that ameliorates colitis in mice, indicating the anti-inflammatory role of IL-19 in the gastrointestinal tract. Interestingly, there were two IL-19 polymorphisms that might have protective roles in patients with UC [23]. Jordan et al. [14] demonstrated that human IL-19 regulates immunity through the auto-induction of IL-19 and the production of IL-10. We also found that the murine IL-19 gene transferred to the colon epithelial cells and the treated IL-19 conditioned medium induced IL-10 production in splenocytes and RAW264.7 (Figure 3). However, Liao et al., showed that murine IL-19 induced the production of IL-6 and TNF-α, which contributed to apoptosis in mouse monocyte [24]. Human IL-19 has two potential N-linked glycosylation sites at positions 56 and 135 that makes multiple IL-19 protein bands around the regions of 35–40 kDa on SDS-PAGE [25]. Accordingly, it may suggest that the different glycosylated forms of IL-19 play distinct roles in immunoregulation. In this study, we showed that IL-19 was an IL-10 inducer and that IL-10 production was stimulated at the early stage of DSS-induced colitis (Figure 4). We also found that IL-19 gene expression in the gastrointestinal tract alleviated gut inflammation through the induction of IL-10, as well as decreased the levels of proinflammatory cytokines, including TNF-α, IL-6, and IL-1β (Figure 8).

Gut inflammation can be induced in mice using chemical compounds such as TNBS, oxazolone, and DSS. In 1990, Okayasu et al., described a model in which mice receiving DSS orally developed acute and chronic colitis resembling UC, but the underlying mechanism remained unclear [26]. Several mechanisms, including toxic effects on the epithelium, increased exposure to luminal antigens via the destruction of mucin content, and altered macrophage function due to ingestion of DSS, were proposed [27,28]. Furthermore, inflammatory infiltrates in the acute phase of DSS-induced colitis consisted predominantly of macrophages, neutrophils, and eosinophils [29]. In parallel to our findings that the *S.C.*/pLJD-CMV-mIL19-treated DSS acute colitis mice had reduced macrophage and neutrophil infiltrations (Figure 7).

Human IBD studies provide evidence that CD is characterized by a Th1-mediated immune response [30,31]. However, Boirivant M. et al., described a murine model induced by oxazolone colitis that resembled UC, having a typical Th2-mediated response through the secretion of IL-13 by natural killer T cells [32]. It is suggested that Th1 characterized CD, whereas UC was characterized by a Th2 type immune response. In this study, our murine model of DSS-induced colitis resembling CD indicated that Th1-cytokines played important roles and were associated with excessive Th1-mediated responses [33]. IL-19 influences the maturation of T cells and alters the balance of Th1/Th2 cells in favor of Th2 [34]. It may suggest that IL-19 can alter the excessive Th1-mediated response by shifting to Th2 in DSS-induced colitis. 

Prebiotics and probiotics hold promises for treating IBD, which have been proven in some studies as prescription drugs [35,36]. The oral administration of different probiotic strains, such as *Lactobacillus* and *Bifidobacterium*, prevents DSS-induced acute colitis [37]. Interestingly, our data showed that the oral administration of either *S. choleraesuis* or *S.C.*/pLJD in DSS acute colitis mice can decrease TNF-α, IL-6, and IL-1β production compared with the saline group (Figure 8). Attenuated salmonella strains can interact with model human epithelia and attenuate the synthesis of inflammatory molecules elicited by diverse proinflammatory stimuli by preventing the nuclear translocation of NF-kappaB [38]. Although we might term attenuated *S. choleraesuis* as a probiotic in DSS-induced colitis due to the decrease in pro-inflammation cytokines production, the modulation of the gut microbiome in DSS-induced colitis mice by *S. choleraesuis* could not be ruled out. Therefore, the limitation of these data is the lack of bacterial sequencing before and after each therapy. Another limitation is the lack of evidence via immunofluorescence microscopy for the presence of the GFP signals in the epithelial cells of mice. Although we prove that *S.C.*/pLJD-CMV-mIL19 can express the IL-19 gene in the colon extracts of the colitis mice, more specific investigations to prove the in vivo colon epithelial expression of IL-19 in these mice will be required in the future.

In conclusion, the present study demonstrates for the first time that IL-19 is a gene therapy agent in a murine model of colitis. The immunoregulatory gene delivered by the attenuated salmonella strain offers the perspective of local modulation of the inflammatory responses. It may provide another choice to ameliorate IBD in the future.

## Figures and Tables

**Figure 1 microorganisms-11-01530-f001:**
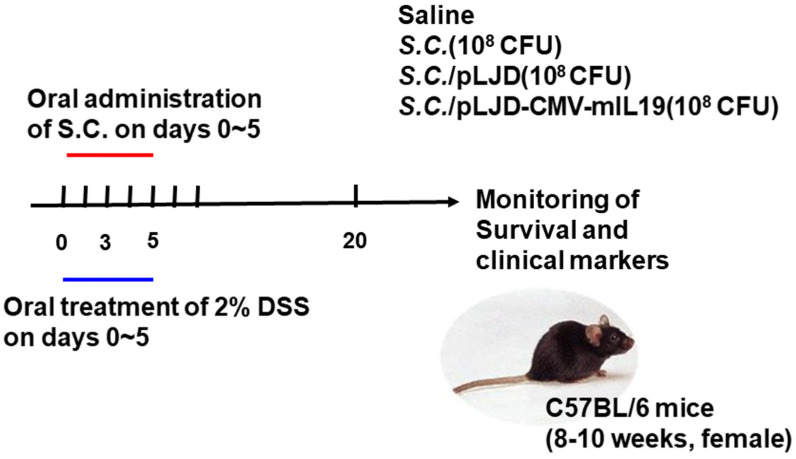
The schedule of gene therapy in DSS–induced colitis mice. The 2% DSS was added to tap water for 6 days and changed to normal tap water at day 6. At day 0, the DSS-treated mice were orally administrated with 10^8^ CFU of *S.C.*, *S.C.*/pLJD, *S.C.*/pLJD-CMV-mIL19 and normal saline (500 μL). The survival and body weight loss were monitored every day. The inflammatory score, diarrhea score, and fecal blood score were monitored at day 3 and day 5.

**Figure 2 microorganisms-11-01530-f002:**
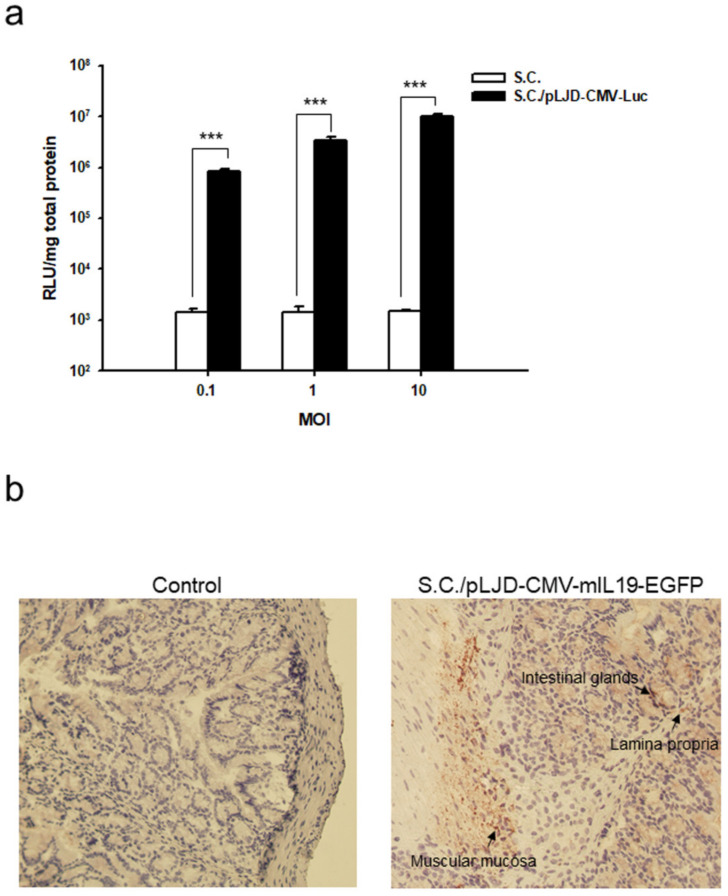
Expression of the reporter genes by *S. choleraesuis* (*S.C.*)-mediated gene delivery in vitro and in vivo. Caco-2 cells were incubated for 8 h with *S.C.*/pLJD-CMV-Luc or S.C. at multiplicity of infections (MOIs) of 0.1, 1, and 10. Monolayers were then washed and cultured in a complete medium for 24 h. (**a**) Luciferase activity (*n* = 3, ***, *p* < 0.001), and (**b**) Immunohistochemical stain of GFP in mice colon specimens after oral administration of 10^8^ colony of infection (CFU) of *S.C.*/pLJD-CMV-mIL19-GFP 24 h later (magnification ×200). Arrows indicate positive signals in the intestinal glands, lamina propria, and muscular mucosa.

**Figure 3 microorganisms-11-01530-f003:**
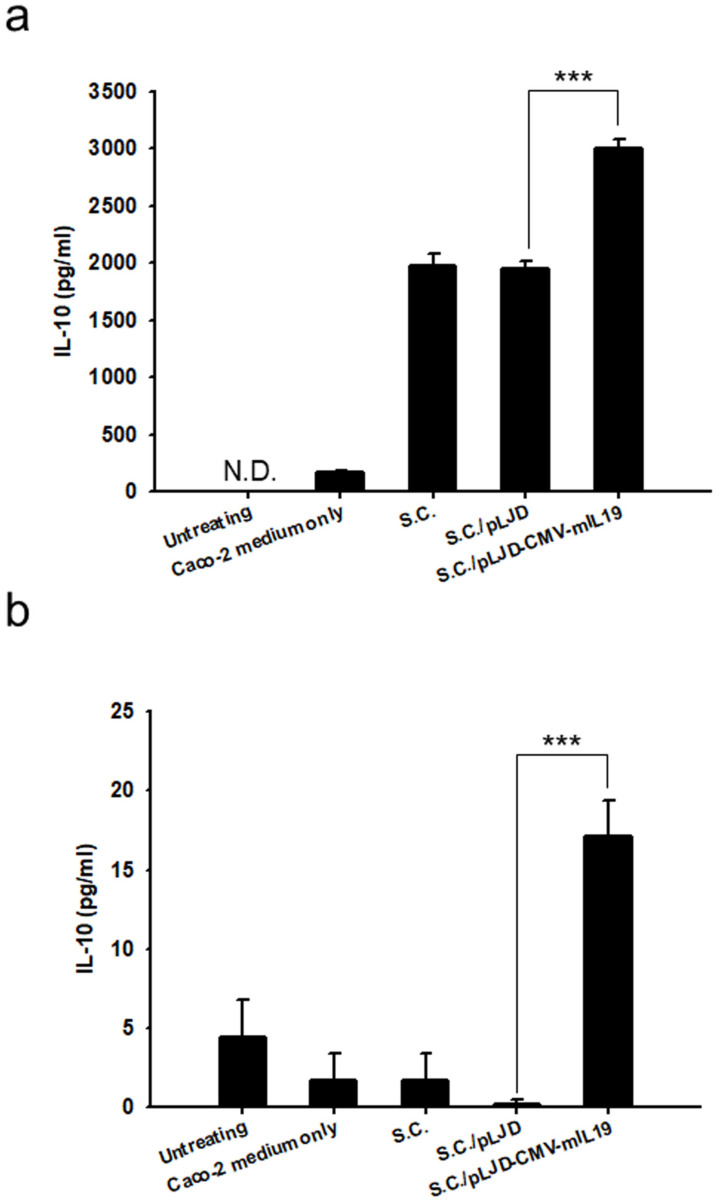
*S.C.*/pLJD-CMV-mIL19 transferred IL-19 to Caco-2 cells and induced IL-10 production in mouse splenocytes and RAW264.7. Caco-2 cells were incubated for 8 h with *S.C.*, *S.C.*/pLJD, and *S.C.*/pLJD-CMV-mIL19. Monolayers were then washed and cultured in complete medium for 24 h. The RAW 264.7 and splenocytes were treated with conditioned medium from Caco-2 for another 24 h. Culture medium was collected, and IL-10 was measured by ELISA in RAW 264.7 (**a**) and splenocytes (**b**) (*n* = 3; ***, *p* < 0.001).

**Figure 4 microorganisms-11-01530-f004:**
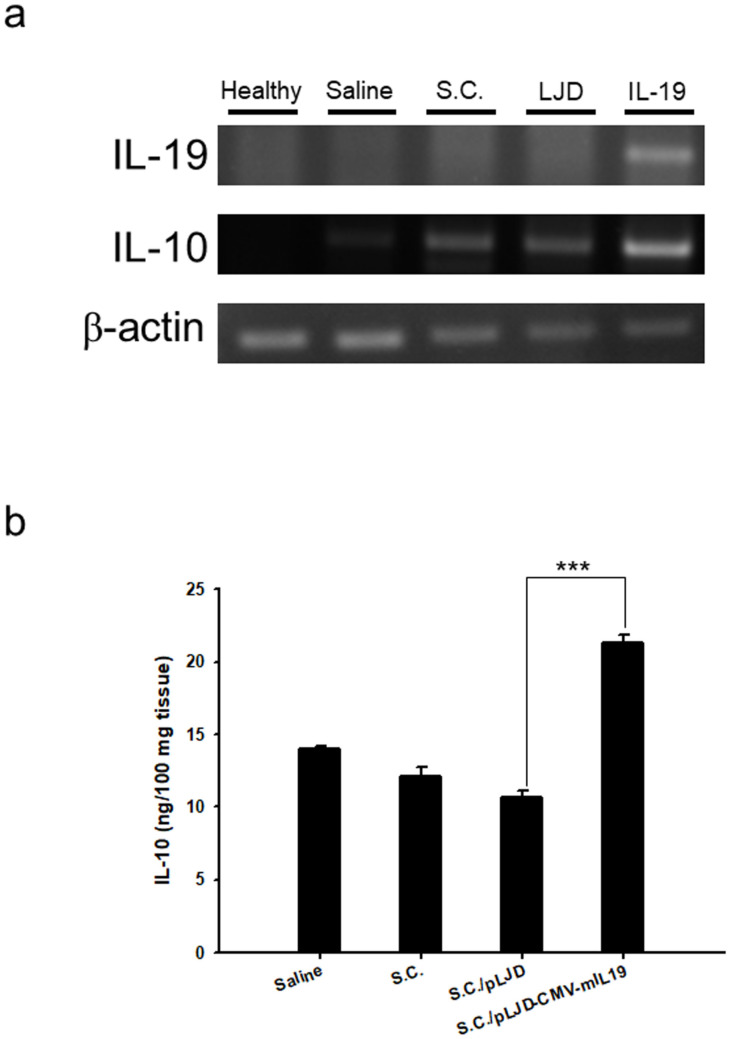
*S.C.*/pLJD-CMV-mIL19- treated mice induced IL-10 production in DSS-induced colitis mice. Disease was induced by 2% DSS for 6 days. 10^8^ CFU of *S.C.*, *S.C.*/pLJD (LJD), *S.C.*/pLJD-CMV-mIL19 treatment, and saline was administered by daily oral administration starting from day 0. At day 3, (**a**) Total RNA was extracted and used to prepare cDNA, and IL-19, IL-10 mRNA transcripts were detected by RT-PCR. (**b**) Colonic homogenates were measured for IL-10 by ELISA in the control and *S.C.*/pLJD-CMV-mIL19-treated DSS colitis mice (*n* = 3, *** *p* < 0.001).

**Figure 5 microorganisms-11-01530-f005:**
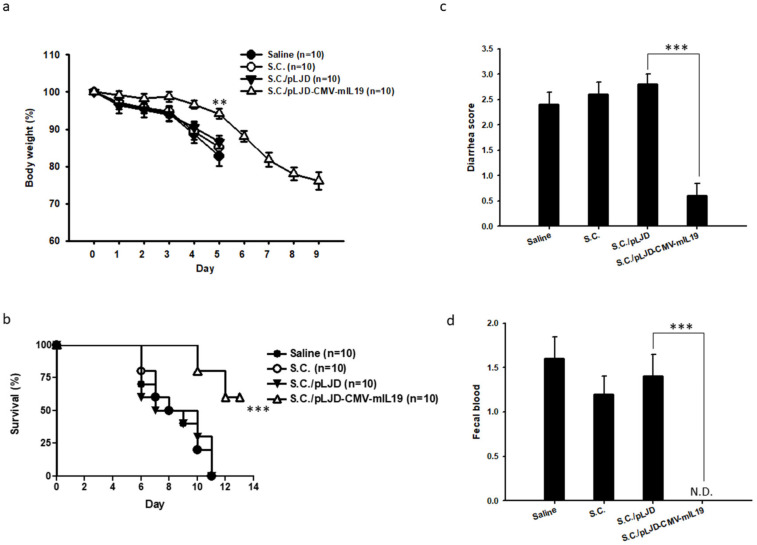
The effects of *S.C.*/pLJD-CMV-mIL19 treatment on the various manifestations of acute DSS colitis in C57BL/6 mice. Disease was induced by 2% DSS for 6 days. 10^8^ CFU of *S.C.*, *S.C.*/pLJD, *S.C.*/LJD-CMV-mIL19 treatment, and saline were orally administered daily starting from day 0. (**a**) body weight and (**b**) survival were monitored daily (** *p* < 0.01; *** *p* < 0.001). At day 5, diarrhea score (**c**) and fecal blood score (**d**) were assessed after DSS acute colitis mice treatment (*n* = 5; ** *p* < 0.01, *** *p* < 0.001).

**Figure 6 microorganisms-11-01530-f006:**
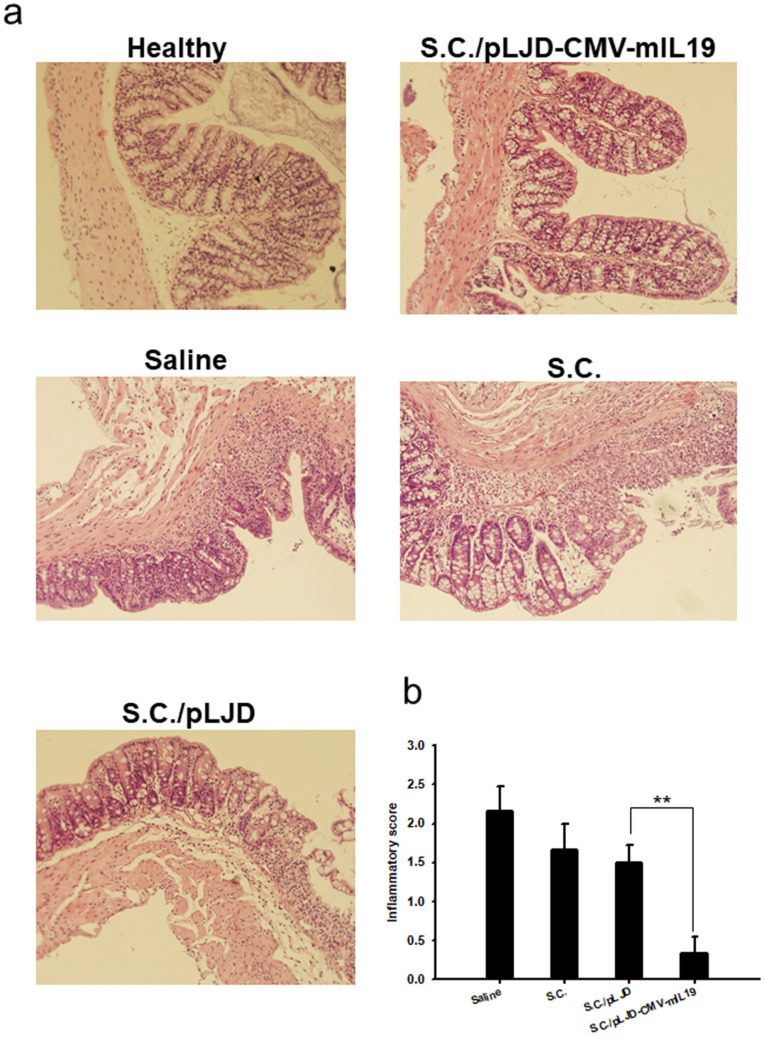
The effect of *S.C.*/pLJD-CMV-mIL19 treatment on the histological appearances of acute DSS colitis in C57BL/6 mice colons. Disease was induced by 2% DSS for 6 days. 10^8^ CFU of *S.C.*, *S.C.*/pLJD, *S.C.*/pLJD-CMV-mIL19 and saline was administered by daily oral administration starting from day 0. (**a**) H&E staining (magnification ×200). (**b**) Inflammatory score was shown at day 5 after treatment (*n* = 6; ** *p* < 0.01). The black arrow indicates loss of crypts and reduction of goblet cells, signs of surface epithelial regeneration, focal ulcerations, and the gray arrow indicates edema in the submucosa, respectively.

**Figure 7 microorganisms-11-01530-f007:**
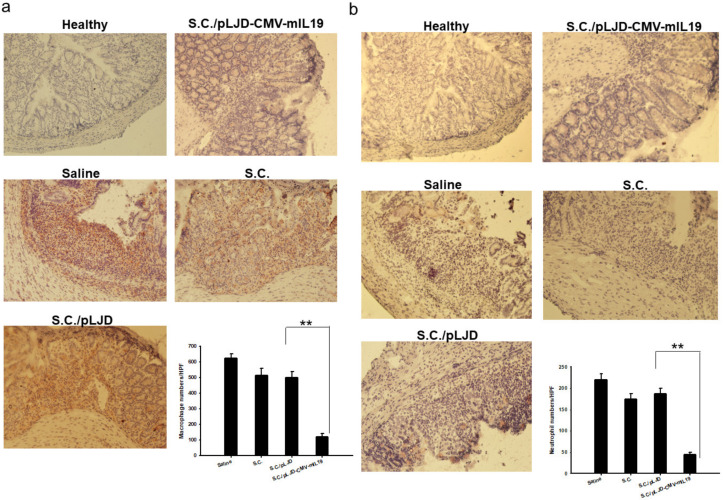
The effect of *S.C.*/pLJD-CMV-mIL19 treatment on the histological manifestations of acute DSS colitis in C57BL/6 mice colons. Disease was induced by 2% DSS for 6 days. 10^8^ CFU of *S.C.*, *S.C.*/pLJD, *S.C.*/pLJD-CMV-mIL19 treatment, /mouse, and saline was administered by daily oral administration starting from day 0. At day 5, the MAC3 (**a**) and Gr1 (**b**) stained cells in the infiltration cells of mucosa were shown (magnification ×200) (*n* = 6; ** *p* < 0.01).

**Figure 8 microorganisms-11-01530-f008:**
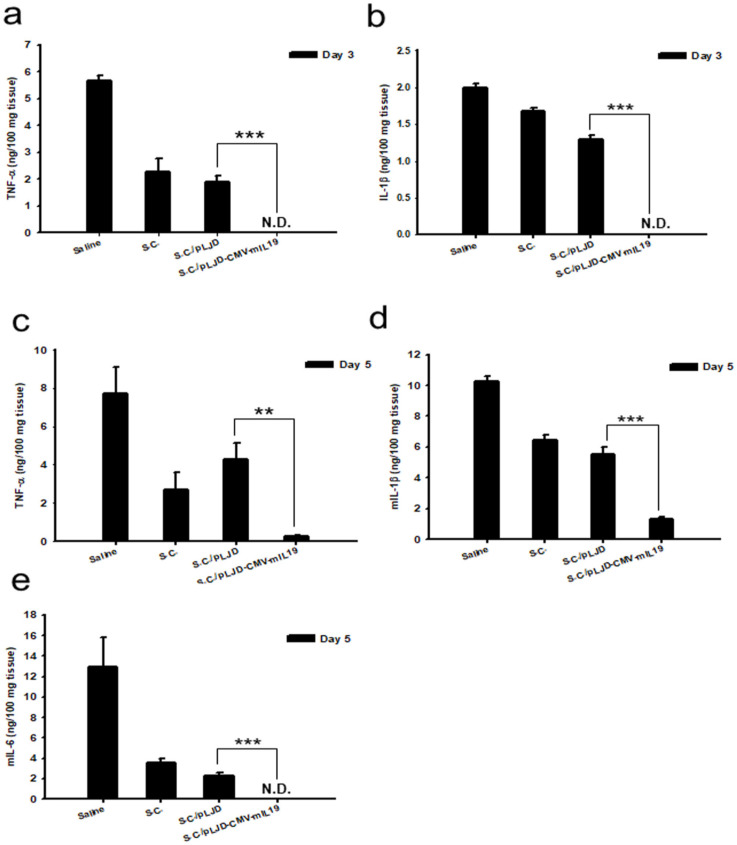
Cytokine production after *S.C.*/pLJD-CMV-mIL19 treatment in C57BL/6 mice induced with acute DSS colitis. Disease was induced by 2% DSS for 6 days. 10^8^ CFU of *S.C.*, *S.C.*/pLJD, *S.C.*/pLJD-CMV-mIL19 treatment and saline were administered by daily oral administration starting from day 0. Expression levels of TNF-α (**a**), IL-1β (**b**) at day 3 and TNF-α (**c**), and IL-1β (**d**), and IL-6 (**e**) at day 5 were measured by ELISA (*n* = 3; ** *p* < 0.01; *** *p* < 0.001).

## Data Availability

The data that support the findings of this study are available on request from the corresponding author.

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
