# Peer review of "Amelioration of Murine Colitis by Attenuated Salmonella choleraesuis Encoding Interleukin-19"

_microorganisms, 2023, doi:10.3390/microorganisms11061530_

Round 1

Reviewer 1 Report

The manuscript demonstrates that S. choleraesuis can be used to transfer an IL-19 vector to both Caco-2 cells and into mice with DSS-induced acute colitis. This process is demonstrated by IHC in the colonic cell line and in the murine colonic tissue, and importantly histologic, physiologic (body weight, fecal blood, diarrhea), and the biochemical measures (cytokine expression) are measured.  The methods employed in the study are sound, as are the statistical methods. The paper uses a previously used treatment methodology and applies it to IFBD. 

The findings are interesting, and the presentation of the data is good. The manuscript should be published. 

Author Response

Thank you very much for taking the time to review our manuscript.

Reviewer 2 Report

Authors Shih-Yao Chen et al. In work entitled Amelioration of murine colitis by attenuated Salmonella choleraesuis encoding interleukin-19, they perform a series of in vitro and animal model tests and highlight how chronic intestinal diseases, again in the animal model, can be modulated by gene therapy through trans acquisition by Eukaryotic cells of bacterial plasmid vectors inducing the expression of Interleukin 19 important modulator of interleukin 10.

The study is fascinating and highlights a long and hard work by the authors. I believe that it can be improved in some aspects and can be published after resolving some major and minor revisions.

Major revisions 

1) please highlight the primary antibody function tests used . In case not were performed I suggest to cite the studies in which such proves were performed and dedicate a few sentences, in materials and methods 

2) figure s1 belongs to the main text, so remove the word supplementary and rename all figures below 

3) indicate for each figure through the use of arrows what the authors would to show and consequently better extend the legend of each figure 

Q

4) in the discussions the corrected sentences  is reported that some studies have observed how Probiotics can modulate IBS. As such in the study bn lacks evidence of bacterial sequencing before and after gene therapy, modulation of the gut microbiome of mice with salmonella modified by the authors cannot be ruled out. In light of this, I urge the authors to highlight this limitation and use cautious attitude in expressing the data throughout the text.

5) There seems to be a lack of evidence by immunofluorescence microscopy of the presence of the plasmid vector in the epithelial cells of mice. I advise the authors to devote a few sentences on this aspect and to prepare such evidence for the next paper as soon as it is published

Minor revisions 

With each acronym used for the first time, insert the scientific term fully.

Example: IL-19......Interleukin-19 

So for each acronym and particularly in the materials and methods section.

Author Response

Comments and Suggestions for Authors

Authors Shih-Yao Chen et al. In work entitled Amelioration of murine colitis by attenuated Salmonella choleraesuis encoding interleukin-19, they perform a series of in vitro and animal model tests and highlight how chronic intestinal diseases, again in the animal model, can be modulated by gene therapy through trans acquisition by Eukaryotic cells of bacterial plasmid vectors inducing the expression of Interleukin 19 important modulator of interleukin 10.

The study is fascinating and highlights a long and hard work by the authors. I believe that it can be improved in some aspects and can be published after resolving some

major and minor revisions.

Response: Thank you very much for taking the time to review our manuscript. We will respond to your valuable points as follows,

Major revisions 

1)please highlight the primary antibody function tests used . In case not were performed I suggest to cite the studies in which such proves were performed and dedicate a few sentences, in materials and methods 

Response: Thank you very much for the valuable suggestion. We have attached the catalog number of each antibody (page 4, lines 126-127), in which you could find the information of the antibody, including their citations.   

 2) figure s1 belongs to the main text, so remove the word supplementary and rename all figures below. 

Response: Thank you very much for the valuable suggestion. We removed the word supplementary and renamed all figures.

3) indicate for each figure through the use of arrows what the authors would to show and consequently better extend the legend of each figure 

Response: Thank you very much for the valuable suggestion. We have included arrows in Figs. 2 and 6 to indicate colon details.

4) in the discussions the corrected sentences is reported that some studies have observed how Probiotics can modulate IBS. As such in the study bn lacks evidence of bacterial sequencing before and after gene therapy, modulation of the gut microbiome of mice with salmonella modified by the authors cannot be ruled out. In light of this, I urge the authors to highlight this limitation and use cautious attitude in expressing the data throughout the text.

 Response: Thank you very much for such critical information. Indeed, we should use cautious attitude in expressing the data throughout the text. Therefore, we modified the sentence regarding Probiotics as follows” Although we might term attenuated S. choleraesuis as a probiotic in DSS-induced colitis due to the decrease in pro-inflammation cytokine production, modulation of the gut microbiome in DSS-induced colitis mice by S. choleraesuis could not be ruled out. Therefore, the limitation of this data is the lack of bacterial sequencing before and after each therapy (Page 14, line 387-392).   

5) There seems to be a lack of evidence by immunofluorescence microscopy of the presence of the plasmid vector in the epithelial cells of mice. I advise the authors to devote a few sentences on this aspect and to prepare such evidence for the next paper as soon as it is published.

Response: Thank you for the suggestion. Indeed, there is a lack of evidence by immunofluorescence microscopy of the presence of the plasmid vector in the epithelial cells of mice. Therefore, we included another limitation as follows” Another limitation is the lack of evidence by immunofluorescence microscopy for the presence of the GFP signals in the epithelial cells of mice. Although we prove that S.C./pLJD-CMV-mIL19 can express IL-19 gene in colon extracts of the colitis mice, more specific investigations to prove in vivo colon epithelial expression of IL-19 in these mice will be required in the future.” (Page 16, lines 392-396).  

Minor revisions 

With each acronym used for the first time, insert the scientific term fully.

Example: IL-19......Interleukin-19 

So for each acronym and particularly in the materials and methods section.

Response: Thank you very much for the suggestion. We have corrected each acronym throughout this manuscript.

Round 2

Reviewer 2 Report

The manuscript is ready to be considered fro publication.